# Prognostic Value of Cardiovascular Biomarkers in COVID-19: A Review

**DOI:** 10.3390/v12050527

**Published:** 2020-05-11

**Authors:** Maryam Aboughdir, Thomas Kirwin, Ashiq Abdul Khader, Brian Wang

**Affiliations:** 1Department of Medicine, Imperial College London, London W12 0NN, UK; m.aboughdir@gmail.com (M.A.); tkirwin05@gmail.com (T.K.); asa415@ic.ac.uk (A.A.K.); 2Department of Medicine, St. George’s University of London, London SW17 0RE, UK; 3College of Medical and Dental Sciences, University of Birmingham, Birmingham B15 2TT, UK

**Keywords:** COVID-19, coronavirus, cardiovascular disease, SARS-CoV-2

## Abstract

In early December 2019, the coronavirus disease (COVID-19) caused by severe acute respiratory syndrome coronavirus 2 (SARS-CoV-2) first emerged in Wuhan, China. As of May 10th, 2020, a total of over 4 million COVID-19 cases and 280,000 deaths have been reported globally, reflecting the raised infectivity and severity of this virus. Amongst hospitalised COVID-19 patients, there is a high prevalence of established cardiovascular disease (CVD). There is evidence showing that COVID-19 may exacerbate cardiovascular risk factors and preexisting CVD or may lead to cardiovascular complications. With intensive care units operating at maximum capacity and such staggering mortality rates reported, it is imperative during this time-sensitive COVID-19 outbreak to identify patients with an increased risk of adverse outcomes and/or myocardial injury. Preliminary findings from COVID-19 studies have shown the association of biomarkers of acute cardiac injury and coagulation with worse prognosis. While these biomarkers are recognised for CVD, there is emerging prospect that they may aid prognosis in COVID-19, especially in patients with cardiovascular comorbidities or risk factors that predispose to worse outcomes. Consequently, the aim of this review is to identify cardiovascular prognostic factors associated with morbidity and mortality in COVID-19 and to highlight considerations for incorporating laboratory testing of biomarkers of cardiovascular performance in COVID-19 to optimise outcomes.

## 1. Introduction

In early December 2019, the coronavirus disease (COVID-19) caused by severe acute respiratory syndrome coronavirus 2 (SARS-CoV-2) first emerged in Wuhan, China [1]. On January 31st, 2020, the World Health Organisation declared COVID-19 a public health emergency of international concern, and on March 11th, 2020, it was finally characterised as a pandemic [2]. As of May 10th^,^ 2020, a total of over 4 million COVID-19 cases and 280,000 deaths have been reported globally, reflecting the raised infectivity and severity of this virus, yet the lack of widespread testing availability means these figures are likely even higher than reported [3]. It is therefore important to predict the risk of morbidity and mortality, especially in vulnerable patients.

SARS-CoV-2 is an enveloped, non-segmented, single-stranded, positive-sense RNA virus belonging to the Coronaviridae family [4]. SARS-CoV-2 is a zoonotic virus not too dissimilar to the SARS-CoV outbreak of 2002 and the Middle East respiratory syndrome coronavirus of 2012 [5,6,7,8]. This novel coronavirus enters cells via binding of the viral surface spike protein to the angiotensin-converting enzyme (ACE) 2 protein [9]. ACE2 is highly expressed in lung alveolar cells, providing the route of entry for the virus [10]. In addition, ACE2 is also widely present in the myocardium, which has raised concerns due to the possibility of direct viral infection of the cardiovascular system [10].

Although COVID-19 patients present primarily with symptoms of respiratory disease and therefore follow a pneumonia-like treatment plan, it is essential that the cardiovascular system is not ignored and to recognise those presenting with early signs of acute myocardial injury. Of the patients hospitalised for COVID-19 thus far, the prevalence of cardiovascular comorbidities has been staggering. Based on early reports, patients with cardiovascular disease (CVD) may represent 25% of those in an intensive care unit (ICU) plus those with hypertension accounting for 58% of patients [11]. Additionally, Zhou et al. found that myocardial injury, defined by raised serum cardiac troponin I (cTnI) levels, in COVID-19 patients was associated with over 50% mortality rate [12]. Furthermore, heart failure was prevalent in 23% of patients presenting with COVID-19, which was also more prevalent amongst patients who died compared to those who survived (51.9% vs. 11.7%) [12]. This demonstrates how essential it is to recognise those presenting with early signs of acute myocardial injury and to initiate a more intensive treatment plan.

Based on these observations, several theories surrounding the interplay between the pathophysiology of COVID-19 and the cardiovascular system have been postulated [13,14]. Namely, COVID-19 may exacerbate cardiovascular risk factors and preexisting CVD or may increase susceptibility for the development of new cardiovascular complications. Alternatively, CVD or myocardial injury may predispose to worse outcomes in COVID-19 patients, which is reflected by a number of studies whereby established CVD is associated with more severe COVID-19, leading to higher morbidity and mortality. 

With such staggering mortality rates reported, it is fundamental during this time-sensitive COVID-19 outbreak to identify patients with an increased risk of adverse CVD outcomes and/or myocardial injury. One may achieve this through laboratory investigations of biomarkers such as cTnI, brain natriuretic peptide (BNP), D-dimers, and fibrinogen—all of which reflect cardiovascular function and are used as diagnostic tools in addition to assessing the risk of CVD in patients [15,16,17,18]. While these biomarkers are recognised for CVD, there is an emerging prospect that they may aid prognosis in COVID-19, especially in patients with cardiovascular comorbidities or risk factors that predispose to worse outcomes. This is crucial as the speed of deterioration of many COVID-19 patients means any early biomarkers indicative of severe morbidity or mortality may then help prevent this rapid deterioration. Consequently, the aim of this review is to identify cardiovascular prognostic factors associated with morbidity and mortality in COVID-19 and to highlight considerations by summarising the evidence for utilising laboratory testing of biomarkers of cardiovascular performance in COVID-19 to optimise outcomes.

## 2. Biomarkers of Myocardial Injury in COVID-19

COVID-19 patients at risk of serious illness and ICU admission tend to be older and to present with similar comorbidities, including heart failure, hypertension, and coronary artery disease [12,19,20]. In a meta-analysis of 8 studies (46,248 COVID-19 patients in total), CVD was reported as the third most prevalent comorbidity in COVID-19 patients (5%, 95% CI 4%–7%), and patients with severe COVID-19 symptoms had a higher risk of CVD (OR 3.42, 95% CI 1.88–6.22) [21]. Whilst these results were limited by significant heterogeneity due to variations in the severity of COVID-19 patients and follow-up period, a similarly high prevalence of CVD in COVID-19 patients (15%) was observed in the study by Huang and colleagues [1,21]. Notably, Yang and Jin state that COVID-19 patients with established CVD are susceptible to more adverse complications—these patients are therefore also at a greater risk of myocardial injury, which mainly manifests as elevated serum cTnI levels [22]. 

cTnI is a gold-standard necrotic biomarker for myocardial risk assessment worldwide [15]. It is released virtually exclusively in the myocardium in the presence of myocardial injury irrespective of the mechanism of insult [15]. Other biomarkers of myocardial injury that are of diagnostic value include creatine kinase-myocardial band (CK-MB) and BNP, which may provide insight into the severity of symptoms in COVID-19. Although they are already established for CVD, results from emerging studies, as discussed below, elucidate the potential role of these biomarkers, particularly cTnI and cardiac troponin T (cTnT), as predictors of prognosis in COVID-19 patients, as shown in Table 1.

The predictive potential of troponin proteins for severe morbidity in COVID-19 patients has been demonstrated. For instance, Huang et al. reported a substantial elevation of cTnI (>28 ph/mL) in 5 out of 41 (12%) COVID-19 patients [1]. All 5 then developed acute myocardial injury, and 4 out of the 5 were admitted into an ICU—this allows the conceptualisation of cTnI as a prognostic tool in other diseases such as COVID-19 [1]. In addition, a meta-analysis of 4 studies with 341 COVID-19 patients reported a significantly higher cTnI mean difference in patients with more severe COVID-19 symptoms compared with patients with non-severe COVID-19 presentation (25.6 ng/L, 95% CI 6.8–44.5 ng/L), although heterogeneity was relatively high, posing a limitation similar to the previously mentioned meta-analysis [23]. Nevertheless, Shi et al. also identified that 82 out of 416 (19.7%) COVID-19 patients presented with myocardial injury, diagnosed by significantly raised serum cTnI levels [19]. Amongst these patients, there was a significantly higher mortality rate of 51.2% compared to a 4.5% mortality rate in those with normal cTnI levels and no myocardial injury, signifying the serious nature of myocardial injury in COVID-19 patients [19]. Importantly, it demonstrates the potential value of cTnI in foreshadowing the outcomes of COVID-19 patients. 

The possible role of cTnT in COVID-19 prognosis is also exemplified by Guo et al., who reported the elevation of cTnT levels in 52 out of 187 (27.8%) hospitalised COVID-19 patients, all of whom developed myocardial injury [20]. In those 52 patients, mortality was a staggering 59.6% compared to 8.9% in those patients with normal serum cTnT levels [20]. Whilst COVID-19 patients with raised cTnT levels and established CVD had an alarming mortality rate of 69.4%, those with raised serum cTnT levels but no history of CVD still had a relatively high mortality rate of 37.5% [20]. This indicates the prognostic value of detecting elevated cTnT levels in all COVID-19 patients, irrespective of the presence of underlying CVD. Conversely, patients with normal serum cTnT levels and established CVD had a much lower mortality rate of 13.3% compared to the 59.6% rate in patients with elevated cTnT levels [20]. 

Interestingly, Guo et al. also observed a significant positive linear correlation between serum cTnT and plasma C-reactive protein (*p* < 0.001), suggesting a link between the severity of inflammation observed in COVID-19 and myocardial injury [20]. Indeed, several myocarditis autopsy findings of inflammatory mononuclear infiltrate in myocardial tissue have been reported in patients with high viral load—these studies also further explore the changes in cardiac inflammatory markers during COVID-19 manifestation [24,25,26]. It is therefore plausible that, through these inflammatory changes, there is an increased risk of myocardial injury, which manifests as elevated serum cTnT levels and consequently leads to more severe symptoms. 

Whilst cTnI and cTnT have demonstrated remarkable potential in predicting COVID-19 outcomes, BNP too has shown some prospect in the prognosis of COVID-19. Guo and colleagues found that raised cTnT levels were significantly associated with elevated serum BNP levels (*p* < 0.001) [20]. They reported that, alongside the gradual elevation of serum cTnT levels, BNP levels likewise progressively increased in COVID-19 patients whose health deteriorated, contrasting the low and stable serum BNP levels in successfully discharged patients [20]. Similarly, a case report presented the cardiac involvement in deterioration of a COVID-19 patient without preexisting CVD, whereby serum levels of BNP (5647 pg/mL), cTnT (0.24 ng/mL), and CK-MB (20.3 ng/mL) were all elevated—this patient was then admitted to the ICU with myocarditis [27]. Moreover, Shi et al. report significantly raised BNP levels in COVID-19 patients with myocardial injury compared to those without (1689 pg/mL vs. 139 pg/mL, *p* < 0.001)—these patients consequently also had a high mortality rate of 51.2% [19]. As such, the aforementioned findings in these studies are groundbreaking as they reflect the prospect of routinely measuring serum BNP levels in COVID-19 patients at admission to reduce mortality and to prevent deterioration where possible. 

In addition to cTnI and BNP, CK-MB may similarly hold prognostic value in COVID-19. In the study by Wang et al., 36 out of 138 (26.1%) COVID-19 patients were admitted to the ICU with severe symptoms, all of whom had significantly elevated serum cTnI and CK-MB levels (*p* = 0.004 and *p* < 0.001, respectively) compared to non-ICU patients [11]. Perhaps this implies that patients with more severe COVID-19 symptoms have adverse outcomes of acute myocardial injury—reflected by the elevation in CK-MB and cTnI levels. Likewise, this study provides insight into the value of CK-MB, along with cTnI, in categorising COVID-19 patients with an increased risk of adverse outcomes and admission to ICU for health deterioration. The value of CK-MB and cTnI in COVID-19 is also exemplified in the study by Zhou et al., whereby a significant association between elevated CK-MB and cTnI levels and in-hospital death was illustrated (*p* = 0.043 and *p* < 0.0001, respectively) [12]. Similarly, Wan et al. found that creatine kinase was significantly higher in COVID-19 patients with severe symptoms compared to those with mild symptoms (*p* = 0.0016) [28]. These studies demonstrate the benefit of utilising CK-MB in determining the patients that require urgent intervention.

Although BNP and CK-MB have evidently demonstrated some prognostic value in COVID-19, it is important to highlight that, in all studies measuring BNP or CK-MB, cTnI was also measured and it provided the same, if not a clearer, link between myocardial injury and COVID-19 outcomes. Additionally, contrasting findings are reported between studies regarding creatine kinase levels and severe COVID-19 presentation. For instance, whilst Wan et al. found creatine kinase to be significantly elevated in COVID-19 patients with severe symptoms, Huang et al. found no significant difference in serum creatine kinase levels between ICU and non-ICU patients (*p* = 0.31) [1,28]. Therefore, more studies that clearly illustrate a conclusive link between CK-MB and BNP and COVID-19 outcomes will provide a better understanding of their prognostic role. It is through the lack of evidence in the literature that one, therefore, postulates cTnI may be a preferred option compared to CK-MB and BNP, mainly due to its high sensitivity in detecting worsening prognosis and myocardial injury in COVID-19 patients.

It is worth noting that raised serum cTnI levels are similarly associated with a higher risk of mortality in other diseases such as pneumonia (odds ratio = 9.5), sepsis (odds ratio = 1.92), chronic obstructive pulmonary disease (hazard ratio = 4.5), and acute respiratory distress syndrome (hazard ratio = 1.6) [29,30,31,32]. Hence, one may logically also predict a correlation between elevated serum cTnI levels and a higher risk of mortality in COVID-19 patients. Studies have clearly illustrated a significant difference in serum cTnI levels between COVID-19 patients who survived and those who died. cTnI levels provide novel insight into a multi-faceted prognostic use of cTnI in other diseases than CVD as it has proven to be a reliable marker of mortality in the previously discussed studies. During a crisis such as the current COVID-19 pandemic, measuring serum cTnI levels may enable healthcare professionals to predict prognosis and to therefore avoid worsening outcomes in vulnerable patients by identifying them at an earlier stage and by providing them with an intensive treatment plan which tackles both the myocardial injury and COVID-19. 

## 3. Vascular biomarkers

### 3.1. Markers of Coagulation

It has been clear from early reports in China that abnormal coagulation is associated with poor prognosis in COVID-19 patients. Tang et al. clearly illustrated this by retrospectively analysing the coagulation parameters of 183 hospitalised COVID-19 patients [37]. Strikingly, 70.14% of non-survivors matched the diagnostic criteria for disseminated intravascular coagulation (DIC) in later stages of the disease (according to the criteria described by the International Society on Thrombosis and Haematosis) [38]. This contrasts with only one (0.6%) survivor meeting the criteria. Hence, abnormal coagulation is a principal factor involved in the deterioration and high mortality seen in COVID-19. However, it is less clear if coagulation parameters, as seen in Table 2, could be used to stratify patients on admission, thus highlighting those more likely to develop severe disease in order to prompt swift intensive treatment. 

Thus far, D-dimer has demonstrated promise in its ability to hold prognostic value in COVID-19 patients. Wang et al. conducted a retrospective single-centre case series including 138 patients with confirmed COVID-19 [11]. Those who were eventually admitted to ICU had significantly increased D-dimer levels (median D-dimer 414 mg/L vs. 166 mg/L, *p* < 0.0001) on admission compared to those who avoided intensive treatment [11]. This finding was substantiated by a smaller retrospective cohort study that found D-dimer levels were four times the upper limit of normal in patients subsequently admitted to the ICU, a level much higher than non-ICU patients (median D-dimer level 2.4 mg/L vs. 0.5 mg/L, *p* = 0.0042, reference range < 0.5 mg/L) [1]. Remarkably, a multi-centre retrospective cohort study of 191 patients demonstrated that, even after multi-variant analysis, an increased D-dimer on admission was highly associated with in-hospital death (OR 18.42, *p* = 0.003) [12]. Furthermore, 81% of those who did not survive had a D-dimer > 1 μg/mL on admission compared to just 24% of those who survived [12]. This striking evidence greatly supports the prognostic ability of D-dimer. However, those still hospitalised at the end of the study were excluded; thus, only those who had died or been discharged during the study period were counted. Therefore, this may have exaggerated the difference between the groups as only those with more severe disease at an earlier stage would be included in the analysis of those who died. 

Nonetheless, in addition to D-dimer being raised on admission, numerous studies from China have demonstrated that, in non-survivors, D-dimer continues to rise throughout the clinical course of the disease [11,37]. This is compared to a low and stable D-dimer in those who survived. Importantly, an increased D-dimer was highly associated with acute myocardial injury, diagnosed via a raised cTnI, which as mentioned previously has been correlated with an increased risk of in-hospital death [20]. Therefore, it is reasonable to suggest that D-dimer has prognostic value when taken on admission and could also be used to highlight patients who are deteriorating. However, the practicalities of such an implementation would need to be considered. For example, whilst Wan et al. found that a raised D-dimer was associated with a more severe disease, the median D-dimer level in the severe cases was still within the normal range on admission [28]. This could present a barrier in confidently triaging patients on their D-dimer level if it is still below the cutoff. Although, in this study of 135 patients, there was only one fatality, a death rate much lower than previously reported. Therefore, the raised yet normal D-dimer could be explained by a relatively well cohort.

Prothrombin time (PT) may also hold some predictive value in COVID-19 patients. Contrasting evidence has surfaced concerning the association of an extended PT with admission to ICU. Whilst a smaller retrospective cohort study found that those who were admitted to ICU had a significantly longer PT on admission (median PT 12.2 s vs. 10.7 s, *p* = 0.012), Wang et al reported no significant difference [1,11]. Although, of the 138 patients included in Wang et al.’s analysis, a large proportion was still hospitalised and not discharged (61.6%) [11]. Therefore, patients who were not admitted to ICU may have deteriorated and subsequently required intensive care; thus, comparing patients by ICU admission may be unreliable in this cohort. Nevertheless, there is good evidence to support that a prolonged PT is associated with in-hospital death. A large multi-centre retrospective cohort study found that a PT over 16 s was greatly associated with in-hospital death (OR 4.62, *p* = 0.019), whilst Tang et al. found that PT time was significantly increased in non-survivors (median PT 15.5 s vs. 13.6 s, *p* < 0.001) [12]. Tang et al. also demonstrated that, from admission, PT continued to rise in those who did not survive, supporting its association with in-hospital death [37]. Like D-dimer, an increased PT was also associated with acute cardiac injury, implying that abnormal coagulation parameters on admission are associated with myocardial injury [20]. However, as previously discussed, the pathology of this injury, whether infarction or myocarditis, is still unclear.

Similar trends have also been noted in platelet counts. There was no difference in the platelet counts on admission of those admitted to ICU [1,11]. However, a reduced platelet count was associated with in-hospital death and cardiac injury. Zhou et al. reported a much lower platelet count in those who died (median platelet count, 165.5 × 10^9^/L vs. 220.0 × 10^9^/L, *p* < 0.001) with 20% of non-survivors having a platelet count less than 100 × 10^9^/L on admission compared to just 1% of those who survived [12]. In addition, those with raised cTnI on admission had a significantly lower platelet count compared to those without cardiac injury (median platelet count, 172 × 10^3^/μL vs. 216 × 10^3^/μL, *p* < 0.001). This further illustrates that abnormal coagulation is associated with cardiac injury in hospitalised COVID-19 patients [19]. 

Lastly, a study of 183 patients found that fibrinogen degradation products (FDP) were also significantly raised on admission in patients that did not survive (median FDP 7.6 μg/mL vs. 4.0 μg/mL, *p* < 0.001) [37]. Whilst fibrinogen levels showed no significant difference on admission, it was significantly lower in non-survivors in late hospitalisation [37]. This suggests that a decreasing fibrinogen level is associated with the progression of the disease; thus, it may aid in the identification of deteriorating patients. 

In light of the striking rate of DIC in patients who did not survive, it has been suggested that the use of heparin in COVID-19 may be beneficial. Therefore, Tang et al. conducted a retrospective analysis of COVID-19 patients and found that the use of low molecular weight heparin was associated with improved prognosis in severe COVID-19 cases with a markedly elevated D-dimer [39]. This further supports the pivotal role that abnormal coagulation plays in the deterioration of COVID-19 patients and how coagulation parameters may help in determining the prognosis of patients. Furthermore, it demonstrates that a raised D-dimer may also aid treatment optimisation in severe cases of COVID-19.

Evidently, coagulation parameters have demonstrated their prognostic potential in COVID-19 patients. However, in all studies that demonstrated an association between COVID-19 and coagulation markers, D-Dimer consistently provided the clearest link to ICU admission and in-hospital death. Additionally, as seen in Table 2, it has been the most widely studied biomarker and thus may be the most reliable in predicting the outcome in COVID-19 patients.

### 3.2. Angiotensin II

As previously mentioned, SARS-CoV-2 uses the ACE2 receptor for entry into target cells, found commonly in the lungs, heart, and vessels [10]. ACE converts angiotensin I to angiotensin II, which can then activate the angiotensin II receptor type 1 [40]. Angiotensin II has profound effects not limited to the cardiovascular system, including vasoconstriction; the release of pro-inflammatory cytokines, such as IL-6; as well as pro-oxidative effects [41,42,43]. Numerous studies have demonstrated that the use of ACE inhibitors (ACEIs) and angiotensin II receptor I blockers (ARBs) lead to an increase in expression of the ACE 2 receptor [40]. This has sparked debate surrounding the use of ACEI/ARBs due to potentially enhancing the risk of infection by increasing the entry way, ACE2.

Interestingly, it has been revealed recently that the plasma levels of Angiotensin II were raised in infected patients compared to that of healthy controls [26]. This may be in part explained by the reduction of ACE2 due to the binding and internalisation of the enzyme caused by the virus. Moreover, the level of angiotensin II in COVID-19 patients was strongly associated with viral load and lung injury, suggesting that COVID-19 was causing an imbalance in the renin–angiotensin system [26]. This implies that angiotensin II may be a mediator of the disease, leading to pulmonary vasoconstriction and inflammatory or oxidative organ damage. Therefore, it would be not unreasonable to suggest that the use of an ACEI or ARB may be beneficial in the treatment of COVID-19. Whilst there have been no other studies to date to our knowledge, this suggests that angiotensin II could be used as a biomarker to stratify patients as those with higher angiotensin II would have increased risk of organ failure, thus requiring more intensive treatment. Furthermore, this could provide an explanation for the increased risk of myocardial injury whilst hospitalised with COVID-19. The powerful vasoconstrictive effects of angiotensin II may increase the demand on the heart whilst potentially inducing oxidative damage. This risk of myocardial injury may then be further increased by the coagulative state mentioned previously. However, further studies are essential to support these findings, especially when considering the small sample size of patients in the study.

## 4. Concluding Remarks

Biomarkers of acute myocardial injury have evidently revealed their potential in predicting worsening prognosis for COVID-19 patients with and without myocardial injury. cTnI provides remarkable prognostic value for patients at increased risk of worsening outcomes and in-hospital mortality, though studies have also shown the association of raised CK-MB and BNP levels with more severe symptoms of COVID-19. Raised serum cTnT and cTnI levels show a clear correlation with deteriorating health and increased mortality in patients with established CVD or cardiovascular risk factors and even in those presenting without a history of CVD. As a result, detecting elevated serum cTnT or cTnI levels on admission as a routine procedure may be invaluable to reduce mortality and severe COVID-19 patients during a time when ICUs are operating at maximum capacity. Additionally, it may allow healthcare professionals to initiate intensive treatment in those vulnerable patients before COVID-19 symptoms worsen. 

Collectively, the evidence presented suggests a common coagulation activation in patients that die from COVID-19. D-dimer has demonstrated predictive value for both ICU treatment and in-hospital death when taken on admission. Furthermore, FDP, PT, and platelets when taken on admission may also highlight those more likely to die in hospital. Therefore, the measurement of coagulation parameters on admission may help in the assignment of scarce ICU beds. The continued activation of coagulation throughout the clinical course of non-survivors, evidenced by an increasing D-dimer level and PT plus a decreasing fibrinogen level, may help identify deteriorating patients that require extra support or palliative care. Furthermore, plasma levels of angiotensin II may offer a novel method of predicting disease severity. Also, the pathogenic role of angiotensin II in COVID-19 and the potential use of ACE/ARBS needs to be more clearly elucidated. 

Nonetheless, when considering the prognostic potential of these biomarkers, it is poignant to contemplate whether they are causative in the deterioration of COVID-19 or simply a consequence of disease progression. Additionally, the mechanism concerning the abnormal biomarker levels should be elucidated. For instance, many of the markers of coagulation are raised in inflammatory or hepatic diseases and, thus, are nonspecific (Table 2). Hence, whilst it is clear that the body is in a pro-coagulative phase, the cause of this is unclear. Further investigation into the role of these biomarkers may permit insight into the pathogenesis of SARS-CoV-2. 

Similarly, the exact pathology of myocardial injury in COVID-19 is unknown, although this review has highlighted possible mechanisms to be explored. Firstly, the high association with abnormal coagulation may suggest a causative link. Alternatively, a common trigger, such as angiotensin II, might instigate both coagulation activation and myocardial injury. Nevertheless, more studies are required to elucidate the specific mechanism of myocardial injury and its association with severe inflammation in COVID-19, along with the subsequent detrimental symptoms that often lead to mortality in vulnerable patients.

When reviewing the literature published thus far on COVID-19, the requirement for multi-centre studies with larger cohorts and clinical power is abundantly clear. Furthermore, due to the high demand for research to published, numerous papers included in the review comprise of patients still not discharged from hospital. Consequently, the data has incomplete endpoints, thus reducing the potential clinical translation of their findings. Moreover, the evidence presented only concerns patients presenting to hospital, and further studies in outpatient, primary care, or community settings are required to get a full overview of the clinical severity and cardiovascular impact.

Whilst these cardiovascular biomarkers present excellent prognostic potential, the implementation of their use should be considered. These are routine blood tests done for many well-resourced hospitals; hence, minimal change in practice would be necessary to swiftly implement their use. However, for countries or hospitals with less clinical resources, implementation may be challenging. Furthermore, those less equipped are those most likely to benefit from a prognostic test that would aid in the assignment of scarce resources. Therefore, the development of a rapid tests which could quickly determine an increase in prominent biomarkers may be extremely beneficial. If this was distributed to countries or hospitals less equipped to treat COVID-19, it could greatly support the global battle against this pandemic.

## Figures and Tables

**Table 1 viruses-12-00527-t001:** A table highlighting the biomarkers of myocardial injury that may have prognostic value in the coronavirus disease (COVID-19).

Cardiac Biomarker	Definition	Association with COVID-19	Prognostic Potential	References
cTn	cTnI and cTnT are gold-standard necrotic biomarkers for myocardial injury irrespective of the mechanism of insult [15].	Raised cTnI/cTnT is associated with • Acute myocardial injury • ICU admission • In-hospital death • Severity of inflammation in COVID-19	+++	[1,19,20,23]
BNP	BNP is a predictor of adverse outcome following acute myocardial injury. BNP concentrations increase immediately following myocardial injury, with the extent of increasing correlating with the injury size [16,33].	Raised BNP is associated with • Acute myocardial injury • ICU admission • In-hospital death	++	[1,20,27]
CK-MB	CK-MB is a biomarker of myocardial damage and reperfusion. Raised CK-MB levels are correlated with injury size and are predictors of poor prognosis [34,35,36].	Raised CK-MB is associated with • Acute myocardial injury • ICU admission • In-hospital death	+	[1,11,12,28]

Raised cTnI/cTnT, BNP, and CK-MB levels are all associated with deteriorating clinical parameters in COVID-19 patients. Prognostic potential as judged by the authors on association with clinical findings and the quality of the literature in support of this. cTn cardiac troponin, cTnI cardiac troponin I, cTnT cardiac troponin T, BNP brain natriuretic peptide, CK-MB creatine kinase-myocardial band, ICU intensive care unit.

**Table 2 viruses-12-00527-t002:** A table highlighting the vascular biomarkers that may have prognostic value in COVID-19.

Vascular Biomarker	Definition	Association with COVID-19	Prognostic Potential	References
D-Dimer	D-dimer is a marker of fibrinolysis. Increased D-Dimer levels are associated with, but not limited to, venous thromboembolism, inflammation, and pregnancy. It is relatively nonspecific [17,44].	Increased D-Dimer is associated with • ICU admission • In-hospital death • Acute myocardial injury	+++	[1,11,12,20,28,37]
PT	PT is used to evaluate the extrinsic and common pathways of coagulation. It is the time taken for plasma to clot after adding thromboplastin. PT is increased in DIC and can be a sign of liver disease or vitamin K deficiency [45].	Increased PT is associated with • In-hospital death • Acute myocardial injury • Potentially associated with ICU admission	++	[1,11,12,20,37]
Platelet Count	Number of platelets in a volume of blood: Decreased in many conditions, namely DIC, anaemia, and marrow failure [46].	Reduced platelet count is associated with • In-hospital death • Acute myocardial injury	++	[11,12,19]
Fibrinogen	Fibrinogen is an acute phase protein involved in platelet aggregation and is decreased acutely by consumption due to DIC or chronically due to hepatic impairment [18].	Decreasing fibrinogen levels correlates with deteriorating clinical parameters in COVID-19 patients.	+	[37]
FDP	FDP are fragments released following plasmin-mediated degradation of fibrinogen/fibrin and raised in inflammatory and thrombotic conditions [18].	Raised FDP is associated with in-hospital death in COVID-19 patients.	+	[37]
Angiotensin II	Angiotensin II is a circulating hormone involved in the renin–angiotensin system. It is a regulator of blood pressure through vasoconstriction and sympathetic nervous stimulation [41].	Angiotensin II is raised in infected patients compared to control. Raised plasma levels of Angiotensin II associated with • Viral load • Lung injury	+	[24]

Raised levels of D-dimers, Prothrombin time (PT), fibrinogen degradation products (FDP), and angiotensin II and reduced platelet count and fibrinogen levels are all associated with deteriorating clinical parameters in COVID-19 patients. Prognostic potential is judged by the authors on association with clinical findings and the quality of the literature in support of this. PT, prothrombin time; FDP, fibrinogen degradation products; DIC, disseminated intravascular coagulation; ICU, intensive care unit.

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
