# Peer review of "Prognostic Value of Cardiovascular Biomarkers in COVID-19: A Review"

_viruses, 2020, doi:10.3390/v12050527_

Round 1

Reviewer 1 Report

In this paper, the authors review the current studies reported in the literature that have shown the importance of specific cardiovascular disease (CVD) biomarkers in COVID-19 prognosis. While there may be many factors associated with COVID-19 morbidity, recent multiple reports suggest CVD to have a strong association with it. This review highlights this point and promotes inclusion of detecting CVD biomarkers as a routine procedure on admission to stratify the COVID-19 patients requiring intensive treatment, which may potentially reduce mortalities.

The article is timely and provides a good overview of related reports, their limitations, as well as directions for future clinical studies. The paper is well-written and I enjoyed reading it. I have only one major comment and few minor comments/suggestions. I would recommend the authors to address these before the paper is considered for publication.

Major comment

My major comment for this paper is to summarise all CVD biomarkers in a table along with the associated reports. This will give a clear idea to the reader about the CVD biomarkers that are being discussed in the review and those interested can then look into the details.

Minor comments

  1. Line 41: I don’t understand why these two papers are cited here. According to the sentence, some introductory papers related to SARS-CoV and MERS-CoV should be cited here.
  2. Lines 60-66: This paragraph seems too speculative without proper references for the statements.
  3. Lines 70-73: Add reference.
  4. Line 98: BNP already defined in the previous section. I found similar definition issues at other places in the paper. Please check and rectify.
  5. Lines 167-178: Good points raised for being cautious about prognostic role of CK-MB and BNP in COVID-19 patients due to literature reports with contrasting findings.
  6. Lines 341-347: I appreciate the authors for mentioning the limitations of their review as well as guidelines for future clinical association studies.
  7. The authors should also comment in the concluding remarks about how easy and how good are the detecting tools for these biomarkers and which one are more reliable and more accessible throughout the world.
  8. It will also be important to add a brief comment in the concluding remarks regarding the correlation or causation relation between these biomarkers and COVID-19. The authors should comment on the correlation or causation relation between the markers as well.

Reviewer 2 Report

The manuscript from Dr Aboughdir et al. provides a summary of recent studies on cardiovascular biomarkers in COVID 19 and explores the prognostic value of those biomarkers for the disease severity by analyzing the recently reported data. In general, this is a well-written and insightful review offering a good update on COVID disease with a focus on CVD markers. I have the following comments to address:

  1. The manuscript could be significantly strengthened by 1-2 tables summarizing the latest findings on cardiac biomarkers and covid severity and mortality.
  2. While discussing the prognostic value of cardiovascular biomarkers, the authors may include a more comprehensive discussion on cardiac inflammatory markers such as CRP, IL6 and IL1b etc
  3. Prothrombin time should be abbreviated as PT. The original reports determined PT not PTT. PTT (partial thrombin time) and PT are different assays.
  4. The interpretation of coagulation abnormality should be more accurate, f.e. an increase in PT does not necessarily reflect cardiac injury, instead, it suggests hepatic injury.
  5. Reference citation should be carefully checked throughout the manuscript, f.e. on page 6 line 241, “Although, at the time of submission of the manuscript…”, it is ambiguous, what does “the manuscript” refer to? The authors’ own review manuscript or their cited reference 1 or ref 9?

Round 2

Reviewer 2 Report

My concerns have been satisfactorily addressed and the manuscript has been thoroughly revised and improved. No further comments.